# Immunotherapy, Radiotherapy, and Hyperthermia: A Combined Therapeutic Approach in Pancreatic Cancer Treatment

**DOI:** 10.3390/cancers10120469

**Published:** 2018-11-28

**Authors:** Javed Mahmood, Hem D. Shukla, Sandrine Soman, Santanu Samanta, Prerna Singh, Shriya Kamlapurkar, Ali Saeed, Neha P. Amin, Zeljko Vujaskovic

**Affiliations:** 1Department of Radiation Oncology, University of Maryland School of Medicine, Baltimore, MD 21201, USA; ssoman@umaryland.edu (S.S.); santanu.samanta@umm.edu (S.S.); prerna.singh@umaryland.edu (P.S.); shriya.kamlapurkar@umaryland.edu (S.K.); AliSaeed@umm.edu (A.S.); Neha.Amin@umm.edu (N.P.A.); ZVujaskovic@som.umaryland.edu (Z.V.); 2Department of Pharmaceutical Sciences, School of Pharmacy, University of Maryland, Baltimore, MD 21201, USA

**Keywords:** immunotherapy, pancreatic cancer, targeted hyperthermia, radiation therapy, tripartite, metastasis, tumor microenvironment

## Abstract

Pancreatic cancer (PC) has the highest mortality rate amongst all other cancers in both men and women, with a one-year relative survival rate of 20%, and a five-year relative survival rate of 8% for all stages of PC combined. The Whipple procedure, or pancreaticoduodenectomy, can increase survival for patients with resectable PC, however, less than 20% of patients are candidates for surgery at time of presentation. Most of the patients are diagnosed with advanced PC, often with regional and distant metastasis. In these advanced cases, chemotherapy and radiation have shown limited tumor control, and PC continues to be refractory to treatment and results in a poor survival outcome. In recent years, there has been intensive research on checkpoint inhibitor immunotherapy for PC, however, PC is characterized with dense stromal tissue and a tumor microenvironment (TME) that is highly immunosuppressive, which makes immunotherapy less effective. Interestingly, when immunotherapy is combined with radiation therapy (RT) and loco-regional hyperthermia (HT), it has demonstrated enhanced tumor responses. HT improves tumor killing via a variety of mechanisms, targeting both the tumor and the TME. Targeted HT raises the temperature of the tumor and surrounding tissues to 42–43 °C and makes the tumor more immunoresponsive. HT can also modulate the immune system of the TME by inducing and synthesizing heat shock proteins (HSP), which also activate an anti-tumor response. It is well known that HT can enhance RT-induced DNA damage in cancer cells and simultaneously help to oxygenate hypoxic regions. Thus, it is envisaged that combined HT and RT might have immunomodulatory effects in the PC-TME, making PC more responsive to immunotherapies. Moreover, the combined tripartite approach of immunotherapy, RT, and HT could reduce the overall toxicity associated with each individual therapy, while concomitantly enhancing the immunotherapeutic effect of overall individual therapies to treat local and metastatic PC. Thus, the use of a tripartite combinatorial approach could be promising and more efficacious than monotherapy or dual therapy to treat and increase the survival of the PC patients.

## 1. Introduction: 

Pancreatic cancer (PC) is a lethal disease with a high mortality rate in both men and women, with a median five-year survival rate of 28–32% for localized disease and 8% for all stages [1,2]. In 2017, there was an alarming rise of PC-related deaths in the United States with a 3.2% increase in new cases [3]. Surgical resection via pancreaticoduodenectomy (Whipple) improves survival, with a median survival of 25 months (for margin negative resections) compared to 11 months for unresectable [4]. Due to a lack of sensitive biomarkers for early diagnosis, the disease is frequently diagnosed in the advanced stages, and less than 20% of diagnosed patients have the option to undergo surgery. Typically, for PC treatment, resectable patients undergo pancreaticoduodenectomy followed by a combination of fractionated radiation therapy (RT) and chemotherapy as adjuvant therapies. Unresectable patients typically undergo concurrent chemotherapy or chemo-RT. In cases of borderline resectability, chemotherapy or chemo-RT may be used neo-adjuvantly (prior to surgery), with the hope of converting the patient to the resectability state. Standard chemotherapeutic agents used to treat PC include nucleoside analogs, such as gemcitabine or a combination drug therapy called FOLFIRINOX (infusional FU/leucovorin, oxaliplatin, and irinotecan). Neoadjuvant and adjuvant tumour-targeted RT is often prescribed using a standard fractionation of 1.8–2 Gray (Gy) daily treatment to a total dose of 45 Gy to regional lymph nodes and 50.4–56 Gy to the gross tumor with concurrent capecitabine anti-neoplastic treatment, which is a fluoropyrimidine carbamate. More recently, an epidermal growth factor receptor (EGFR) inhibitor, erlotinib, has been implemented as a treatment modality for PC. In a randomized clinical trial, erlotinib, when combined with gemcitabine, proved to be more effective in disease control and overall survival than gemcitabine alone when used in mutated EGFR metastatic patients [5]. Nonetheless, PC shows a limited response to these individual therapies, making it relatively non-responsive to treatment [6]. PC is more resistant to chemotherapy as well as to other standard treatment modalities compared to other solid tumors [7,8]. Furthermore, the unique feature of PC is the presence of an abundant fibrous stroma, which acts as a physical barrier to restrict intra-tumoral cytotoxic drug infiltration and creates a hypoxic microenvironment that decreases the efficacy of radiotherapy as well. PC is characterized by duct-like, tubular, highly fibrotic stromal growth, which is composed of extracellular matrix (ECM), stromal fibroblasts, various immunosuppressive cells, and a complex extracellular matrix comprised of glycosaminoglycans, proteoglycans, and collagens. Hyaluronan (HA) is one of the predominant glycosaminoglycans present in the PC stroma. Recently, PEGylated human recombinant hyaluronidase combined with gemcitabine tried to overcome the negative stromal environment, but ultimately failed to show benefits, highlighting the challenge to overcome the negative tumor microenvironment (TME) [9]. PC stroma also creates a leaky microvasculature with an increased intra-tumoral interstitial fluid pressure (IFP) that is responsible for creating a hypoxic environment in the pancreatic tumor [10]. Hence, the high rate of deaths among PC patients is largely due to lack of effective therapies [11].

As highlighted above, PC treatment per established guidelines makes use of multi-modality therapy (in the form of chemotherapy, RT, and/or surgery). Although multi-modality treatment has improved outcomes, survival remains dismal. This highlights the need for additional modalities, which operate via different mechanisms and that ultimately converge to promote tumor killing and cancer eradication. Additional novel modalities must target not only the intrinsic cancer cell, but also the TME. In this review, we highlight the emerging roles of immunotherapy (IT) and hyperthermia (HT) as additional modalities in the PC treatment armament and we describe how the combination of IT, HT, and RT through distinct and overlapping mechanisms target both the tumor cell and the TME.

Monotherapy can cause resistance to therapeutic modalities that can be immunologically classified as primary, adaptive, or acquired, and involve an innate refractory response or a development of an avoidance mechanism by the tumor or TME. Recent investigations have revealed that immunotherapy when used in combination with other modalities show promising therapeutic outcomes for a variety of malignancies [12], however, immunotherapy did not show any overwhelming effect in PC treatment outcome [13]. The use of multiple therapies has the benefit of targeting multiple pathways that can ultimately lead to cancer cell death, and it also diminishes the chances of developing resistance. Following this general trend, many currently ongoing and completed clinical trials in PC are pairing a targeted agent, such as immunotherapy, with the mainstay of chemoradiation therapy. The layering of such therapies converges to modulate the PC-TME immunogenically. One such therapy, RT, has the capability alone to trigger immunogenic, cell-specific tumor recognition through neo-antigen release, which can potentiate cancer cell death and stimulate an adjuvant immune effect. A less commonly used treatment modality to consider for amplification of this immune effect is targeted hyperthermia (HT), which could amplify this immune response through increased tumor oxygenation, perfusion, and lymph node dendritic cell trafficking [14]. Further, combining HT with RT and immunotherapy as a tripartite modality has shown promise to treat PC in a syngeneic preclinical model of PC (unpublished data). Since PC is most often detected at advanced stages, immune based approaches combined with RT and hyperthermia are gaining importance [14,15,16,17,18]. In the current review, we highlight the effectiveness of RT, HT and immunotherapeutic drugs as a tripartite modality to treat PC.

## 2. Pancreatic Cancer Tumor Microenvironment (PC-TME)

PC is characterized by a heterogeneous and immunosuppressive environment and consequently poses an enormous challenge to effective immunotherapy. The PC-TME is predominantly populated by Myeloid Derived Suppressor cells (MDSCs), macrophages, and regulatory T cells (Tregs) during early stages of PC tumorigenesis [19]. The presence of abnormally high stromal desmoplasia produces a distinctive TME, which is also dominated by pancreatic stellate cells, fibroblasts, ECM, and high Treg/Teffs ratio (T regulatory cells/inactivated effector T cells ratio). Adversely, PC tumors can develop adaptive environments, which can alter anti-tumor T cell responses. This alteration in T cell accumulation in the tumor, and their activation in the intra-tumoral environment leads to T cell exhaustion during immune response activation in TME. In addition, interferon gamma (INF-γ) secreted by T cells triggers programmed death-ligand 1 (PD-L1) upregulation and produces a PD-L1 signal which helps cancer cells to maintain an immunosuppressive environment [20]. It has also been observed that widespread MDSC intrusion in pancreatic cancer adenocarcinoma (PDAC) may lead to aberrantly functional infiltrating T cells [21], which helps in creating immunosuppressive TME.

Furthermore, as the PC progresses to advanced stages, the number of Tregs and Teffs contribute to an immunosuppressive environment in cancer tissues while stroma can block normal effector T cell populations [22]. Recent investigations also suggest that a lack of recognition by T-cells of tumor antigens or cancer cells can be due to a mechanism to avoid presenting them on the surface, most notably by major histocompatibility complex I (MHC I) downregulation [12]. In addition, current reports also suggest introduction of a mutation in INF-γ receptor 1&2 genes, which gives an advantage to the tumor cells to escape from CD8+ T-cells and avoid an antitumor response, and may also contribute to create an immune suppressive TME. It is through these alterations that tumors can develop resistance to immunotherapy [12,23]. Further, two phenotypes are categorized based on the degree of immune infiltration of T-lymphocytes and this is often classified as ‘cold’ and ‘hot’ tumor [24]. In hot tumors, there seems to be abnormal variation in CD8+ and regulatory T cells, which is responsive to immunotherapeutic drugs [25]. On the contrary, the cold tumor phenotypes are predominant in the early stages of tumorigenesis, which are refractory to immunotherapy, and have shown only a 20–40% response to immune checkpoint inhibitors [26].

## 3. Specific and Integrated Immunotherapy

### 3.1. Immunotherapy for Pancreatic Cancer employing Checkpoint Inhibitors

Immunotherapy success is based on the targeting of factors specific to cancer cells that can aid in activation or suppression of the immune system. This can be achieved by employing monoclonal antibodies, immune checkpoint inhibitors, cancer vaccines, and immune stimulators, such as cytokines, interleukins, or interferons [27,28,29,30]. PC immunotherapy employing immune checkpoint inhibitors has demonstrated little success in treating advanced stage PC [31]. Effectiveness of many checkpoint inhibitors has been limited to a minority of patients, with up to 60% of patients showing primary resistance to (programmed cell death protein 1) PD-1 [32].

Immunotherapy precisely stimulates the host immune system to fight cancer progression and this novel approach of adding HT and RT could be an attractive modality for PC, where an immunosuppressive environment thwarts the anti-tumor response mounted by chemo and radiation therapies [33]. The PC microenvironment hosts an arsenal of pro-tumorigenic immune cells, such as T-regs and MDSCs, which predominantly maintain an immunosuppressive microenvironment. In PC immunotherapy, there are several therapeutic agents that have been developed, including monoclonal antibodies (mAb), which block the immune checkpoint cytotoxic T-lymphocyte-associated molecule-4 (CTLA-4), and most notably, PD-1 [6,34]. Recently, few other checkpoint inhibitors, such as PD-1 and its ligand, PD-L1, have shown the least success in treating PDAC patients [30]. However, because of the immuno-suppressive TME, a single therapeutic approach has not shown promise in the treatment of PC [27]. Moreover, immunosuppressive regulatory cells, such as dendritic cells and regulatory T cells, exhibit an anti-immunogenic effect in PC and can lead to a weak anti-tumor response, thereby limiting the effectiveness of immunotherapies as an individual approach for the treatment of PC [35]. Recently, targeting of PC specific antigens, including small molecules specific to PC, along with dysregulated checkpoints have been an attractive target.

Small molecule inhibitors against aberrantly regulated receptor tyrosine kinases (RTKs), and focal adhesion kinases (FAKs), when complemented with immune checkpoint inhibitors, have significantly improved immunotherapy and the TME response [36]. The overexpression and amplification of factors, like vascular endothelial growth factor (VEGF), tumor necrosis factor alpha (TNF-α), and transforming growth factor beta (TGF-β), also aid in cancer progression by inhibiting infiltration of macrophages, natural killer (NK) cells, and neutrophils into the TME to mount an anti-tumor immune response [37]. Moreover, TGF-β can block cytotoxic T-lymphocytes and promote collagen synthesis, leading to ECM stiffening. This pronounced TME creates an immunosuppressive environment and interacts with cancer stem cells and tumor stromal cells, making it challenging for better therapeutic outcomes [36]. A prime indicator of the switch to an immune responsive TME is the infiltration and presence of cytotoxic T-cells.

Cytotoxic (CD8a+) T-cells have an ability to selectively distinguish and kill pathogens or unhealthy cells by orchestrating a coordinated immune response, including innate and adaptive immune defences [38]. Many checkpoints safeguard cells of the immune system to ensure that they are not erroneously destroying healthy cells during an immune attack. However, cancer cells have adapted to exploit these immune checkpoints as way to evade immune detection and elimination. By blocking immune checkpoints, including PD-1, PD-L1, and CTLA-4, with mAbs, the immune system can overcome the ability of cancer cells to oppose the immune responses, allowing for destruction of cancer cells [39]. There has been very limited or no success in treating PC employing checkpoint inhibitors [40]. PD-L1 exerts an inhibitory effect on T-cells by binding to the T-cell co-receptor, PD-1, and inhibiting the T-cells’ pathway from instigating an immune response against cancer cells. PD-1 are also present in cancer cells, which are inhibited by anti PD-1 antibodies, and this allows the immune system to destroy the cancer cells. In the metastatic stage, PC has shown resistance to several therapies, including immune checkpoint inhibitors’ monotherapy, which could be attributed to T-cell exhaustion and malfunction [41].

In genetically engineered mouse models, it has been observed that the immunosuppressive TME may interfere with intra-tumoral T-cell activation and turnover. Consequently, it is envisaged that T-cell based immunotherapies might contribute to resistance to checkpoint barriers [42]. Nevertheless, anti-CTLA-4 and anti-PD-1 have demonstrated a powerful anti-tumor response by activating the immune system. Interestingly, both checkpoint inhibitors act spontaneously via two distinct pathways. Anti-CTLA-4 normally blocks the CD28+ co-stimulation by weakening the ligand on antigen presenting cells (APC). On the contrary, anti-PD-1 inhibits signaling pathways mediated by T-cell receptors by binding to PD-L1 and PD-L2, which are present on tumor cells [12]. Pre-clinical observations in murine models and clinical trials have recently assessed anti-PD-1/PD-L1 based therapies against many types of lethal cancers. It has been predicted that a PD-L1 blockade can efficiently block pre-established PC in a mouse model by enhanced IFN-γ production and decreased IL-10 release [42]. Furthermore, it has also been observed in several studies that PC patients gradually develop resistance to PD-1/PDL-1 based immunotherapies, and further investigations on the fundamental cause of this therapeutic failure would contribute to a better design of combined therapeutic approaches [6]. It is further anticipated that an amalgamation of therapeutic approaches could bypass resistance to anti-PD-1/PD-L1 immunotherapy in PC and could facilitate the transition of tumors from immunologically nonresponsive to a responsive mode. Recent investigations have shown that a PD-1 and PD-L1 blockage combined with RT, chemotherapy, and other targeted therapies could synergistically enhance the immune response against PC. Overall, single checkpoint inhibitors are ineffective in PC therapy, underscoring the challenges to make immunotherapy more effective, including overcoming the poor antigenicity, a dense desmoplastic stroma, and a largely immunosuppressive TME.

### 3.2. Immune-Modulation by targeted Hyperthermia as a Treatment Modality for Pancreatic Cancer

Hyperthermia (HT) has been employed as a beneficial modality in treating human cancer. Elevated body temperature has been linked to activate the immune system against tumor cells. Normally, HT treatment is provided by heating the targeted tumor site, increasing its temperature to between 39–43 °C for 30–60 min. Interestingly, during HT, normal tissue has the ability to endure the mild temperature while cancer cells are sensitive to temperature stress and succumb to death [43]. Tumor-targeted HT induces and overexpresses a variety of heat shock proteins (HSPs), which can potentially increase tumor antigenicity [11]. For HT therapy, the tumor is heated up to between 39 to 43 °C, and the elevated temperature can alter the pathophysiology of the cancer cells by enhancing oxygenation of the surrounding TME and blood flow to the tumor [44]. It is well known that HSPs play an active role in antigen recognition in dendritic cells, and that transporting HSP bound antigenic peptides to MHC I molecules results in the triggering of antigen-specific T-cell activation [45,46,47,48]. A well-known HSP60 chaperone protein is involved in antigen dependent T-cell activation that allows IFN-γ secretion and T-cell activation [49]. Interestingly, it has also been observed that HT also upregulates HSP70 chaperone protein, which could play an important role in provoking an immune response against tumor propagation. It is also evident that HSPs play an important role as carriers of tumor derived antigens and could serve to activate antitumor immune responses [50,51]. Thus, further investigations on the key roles of HSPs are required to precisely explore their synergistic role in HT, and their role in antigen presentation among T-cells and dendritic cells during HT treatment (Figure 1) [44]. Recent preclinical investigations have also demonstrated that HT stimulates cytokine production by enhancing T-cell membrane flexibility [52]. In a recent preclinical model, it has been demonstrated that T-cell membrane fluidity leads to an organized interaction between T-cells and APCs, which triggers important signaling pathways at the junction of APC and NK cells [53]. In addition, the hyperthermal effect increases permeability of tumor cells, leading to enhanced immunotherapy drug diffusion, simultaneously alleviating drug resistance in tumor tissues and inhibiting DNA repair in PC patients following RT [54]. Moreover, this therapy has the potential to avoid drug resistance due to a higher blood flow in tumor tissues that could result in a relative increase in immunotherapeutic drug concentration within the tumor (Figure 1).

Recent investigations on the role of HT have shown an effective anti-tumor response against PC [55]. The treatment of cancer with loco-regional HT along with other therapeutic modalities has been shown to boost anti-tumor responses (Table 1) [44,56]. Whole-body HT can enhance lymphocyte and endothelial interaction, which allows increased transport of immune effector cells, like inflammatory neutrophils and lymphocytes, to the TME [57]. In a preclinical model, it has been shown that whole body HT releases HSPs and chemokines, which might hyperactivate the immune responses [58]. However, whole body HT has a significantly adverse impact on other vital organs in the body. Treatment of PC with localized HT is difficult because of its location and the proximity to surrounding organs. A recent clinical trial on localized HT has shown a significant improvement in response in 39 advanced stage PC patients when used with gemcitabine and RT [59]. There are promising results in a preclinical model of PC immunotherapy due to advancements in the delivery method of localized HT. In another clinical trial, intra-abdominal HT has been used in combination with gemcitabine and improved survival of PC patients as well as decreased recurrence of disease [60]. More investigations are underway to elucidate the molecular mechanism of HT induced activation of the immune system and its anti-tumor effect [44].

Recent clinical investigations have suggested that loco-regional hyperthermia has the inherent capability to deliver relatively higher doses of immunotherapeutic drugs to the tumor site. Furthermore, hyperthermia enhances the blood supply to the tumor sites, and simultaneously has an inhibitory impact on the NF-κB pathway, a commonly mutated pathway in PC, making the tumor more susceptible to drugs [61,62]. Furthermore, loco-regional hyperthermia induces HSPs and chaperones that seem to be responsible for blocking the NF-κB pathway, which could also have an anti-tumor effect [50].

### 3.3. PC Immunotherapy with RT as a Treatment Modality

It is known that cytotoxic RT induces DNA damage of cancer cells, and that PC can develop resistance to RT due to TME hypoxia, and the increased potential of cells in S-phase to repair DNA damage [41,63]. Due to a lack of biomarkers for early diagnosis of PC, the present standard of care has shown very limited success. Recent investigations have shown that the presence of a recalcitrant TME and its immunosuppressive effect makes PC more refractory to chemo-radiation therapy, and limited success to immunotherapy [64]. Nevertheless, when immunotherapy is administered in combination with RT, it has shown clinical success. Remarkably, the amalgamated approach of radiotherapy and immunotherapy has produced encouraging outcomes in animal models in various types of cancer. The combined treatment approach could abate individual toxic effects and enhance antitumor efficacy during combined therapy [65]. When immune checkpoint inhibitors, such as anti-CTLA-4 and anti-PD-1, are combined with RT, this stimulates release of tumor antigens, which activates the immune response and mounts a coherent immune assault against the tumor. However, due to limited clinical case studies, the dose and sequence of these modalities further needs to be optimized and their toxic effects studied more precisely [65]. The elucidation of the ideal balance of doses and sequence can be optimized to alter the immunosuppressive PC-TME to an immunostimulatory one, with few to no toxic effects.

RT has shown promising therapeutic outcomes as a combined therapy to treat solid tumors in a preclinical and clinical model (Table 2), and has demonstrated immune stimulating effects that could recharge the immune response, stimulate antigen production, and trigger an abscopal effect [65,66,67]. Further, recent reports also suggest that localized RT could effectively treat unresponsive tumors, which were non-responsive to the anti-CTLA-4 antibody, ipilimumab [31,68,69]. Recent investigations have also demonstrated that RT could activate the immune system and induce immunogenic cell death, counter act against an immunosuppressive TME, and activate treated cells as a vaccine against the tumor [45,49,70]. Thus, it is envisaged that RT could induce an abscopal effect at the non-irradiated site, and limit the spread of localized tumors [26]. Furthermore, recent reports have shown that a combination of immunotherapy and fractionated doses of RT and timing have potential in clinical investigation against many cancer types [65]. However, further investigations are needed to optimize the sequence, dose, and timing of combined therapy to make it a clinically promising modality [71].

### 3.4. Employing Tripartite Modalities, including Radiation, Hyperthermia, and Immune Checkpoint Inhibitors, to Treat Pancreatic Cancer

In recent years, immunotherapy has evolved as an immensely powerful treatment for controlling tumors that were aggressive and intractable to conventional treatment. The advancement in tumor biology understanding and its cross talk with the immune system and its checkpoints have enhanced our understanding in designing effective immune checkpoint inhibitors. Over time, it has been observed that the effectiveness of immune checkpoint inhibitors alone is inadequate in PC. Therefore, further investigations are needed to develop a combined therapy, which could be more effective and enhance the overall immunotherapeutic effects.

Checkpoint inhibitor-based immunotherapy has had partial success against PC and has shown limited effectiveness in a small subset of PC patients [72]. However, to further enhance this treatment, the anti-tumor response of the immune checkpoint inhibitors could be significantly enhanced when used in combination with synergistic modalities, such as RT and hyperthermia treatment [44]. Recent reports have demonstrated that a combination of anti-PD-1/PD-L1 along with complementary checkpoint inhibitors with fractionated doses of RT and HT at 42 °C have shown a promising capability to mount an anti-tumor response to destroy PC cells more efficiently [73] (Figure 2). Hyperthermia releases of HSPs, enhancing the effect of IL-2 in activating T-lymphocytes, increases recruitment of tumor killing immune cells, like natural killer cells, macrophages, and cytotoxic and helper T-cells, inside the TME. When HT is combined with RT, the anti-tumor immune response could be augmented. Furthermore, the lack of clinical efficacy of PD-L1 blockade in PC patients suggests that it may be necessary to address the immunosuppressive effects by immune co-stimulatory agents (anti-CD134), hypofractionated RT, or HT. This tripartite treatment of RT, HT and immunotherapy may overcome PC’s immunosuppressive TME leading to enchanced tumor regression. The tripartite treatment will lead to the release of tumor antigens, recruitment of immune cells, and improvement of the immunosuppression by chemotherapy in the TME, thus improving drug delivery by breaking the stromal barrier [55]. Thus far, there has been a lack of clinical trial data on a tripartite modality-based immunotherapy to treat PC, however, there exists an ample amount of in vitro and preclinical data. For deadly cancer like PC, tripartite treatment offers an exciting and promising new therapy. If future clinical trials can show improved tumor control with relatively mild side effects, then tripartite treatment may receive expedited approval from the European Medicines Agency (EMA) or the Food and Drug Association (FDA).

Recently, in a preclinical model, it has been shown that RT could induce initiation of T-cell assault to exogenous mutated cancer antigens expressed by tumor cells [74]. Thus, the T-cell response against cancer antigens has shown that RT induces a specific immunogenic cell death [75,76,77]. Further, it has also been observed that RT induced T-cell priming against immunogenic cancer cells is mainly mediated by CD8+ and CD103+ tumor infiltrating specific dendritic cells activated by IFN-β production by the STING (stimulator of interferon genes) pathway. Further, there is some evidence that demonstates the activation of the STING pathway by RT [78]. It can be envisaged that recalcitrant tumors not responding to immunotherapy may respond to treatment when combined with RT [26]. These investigations have also delineated that the irradiated cancer cells express MHC class I molecules on its surface in a dose dependent manner, which could be recognized by cytotoxic T lymphocytes [79]. Further, the immune activation of tumor associated antigens could alter the TME by recruiting and activating synergistic radiation induced immune cells. These immune cells can enhance immune cell infiltration at local and global sites, leading to increased tumor cell death [80]. However, despite the positive response evoked by a combination of RT and immunotherapy, a major hindrance in successful immunotherapy against PC is a TME dominated by immunosuppressive cells, such as tumor associated macrophages (TAMs) and MDSCs. In an animal model of PC immunotherapy, when anti-CTLA-4 and anti-PD-1 antibodies were combined with RT, the overall survival was significantly enhanced as compared to treatment without RT [81]. Further, in a mouse model of PC, when RT was combined with colony stimulating factor-1 receptor (CSF) inhibitor, it demonstrated a significantly higher survival [82]. Recently, the use of anti-CTLA4 and/or anti-PD-L1 in combination with RT is being investigated in patients with unresectable and non-metastatic PC. Thus, in a number of preclinical models, combination therapy using RT and immunotherapy has shown promise in metastatic PC treatment [83]. Moreover, the clinical presentations of this combined strategy are under clinical investigations, which may hold great promise to treat PC (Table 2). Additional clinical studies are needed to optimize the RT dose, its fractionation, and combination with immunotherapy agents to achieve maximum therapeutic benefits. 

## 4. Vaccines against Pancreatic Cancer Neoantigens, and its Success in Immunotherapy

PDAC is characterized by an immunosuppressive TME and inflammation, which supports poor clinical outcomes. Identification of molecular targets in PC treatment has been very limited due to immense molecular heterogeneity within the tumor as well as a marked lack of tumor specific antigens. Few tumor antigens have been identified in PC that can be utilized in stimulating an anti-tumor immune response. PC creates an immunosuppressive environment due to the presence of an abnormal load of mutated PC antigens, such as KRAS, CDKN2A, ERBB2, and TP53 [84]. The Cancer Genome Atlas (TCGA) genomic analysis has shown that KRAS, SMAD4, TP53, and CDKN2A genes are highly mutated and specifically KRAS remains to be an elusive therapeutic target in PC. TCGA genomic data has shown that KRAS is one of the driver oncogenes in PDAC and it is 94% mutated at codon 12. Thus, the mutated form of KRAS is predominantly expressed on the cell surface, and has been one of the hindrances in developing successful therapy against PC [4]. Several investigations are underway to design potential KRAS inhibitors, which could be combined with immunotherapy for PC treatment [4,85]. Recent clinical studies in long term surviving PC patients has demonstrated that high quality neoantigens may be the target for T-cells [86]. It has been proposed that these neoantigens are responsible for modulating immunogenicity, clonal variation, and immunoselection during tumor evolution [4]. Comprehensive genomic and proteomic analysis have shown that multiple pathways are operational, which seems to be involved in creating an immunosuppressive environment and poor clinical outcome. Thus, in the clinical setting, immunotherapy as a monotherapy has not been very successful in PDAC [39]. Interestingly, tumor specific antigens (TSA), also called neoantigens, are only expressed on cancer cells and not expressed in normal cells, which could be the easy target [4].

TSA are interesting candidates as targets for PC immunotherapy [87,88] and could be possibly used to treat numerous PDAC patients. Promising tumor associated antigens, including ERBB/HER receptors and mesothelin, are being clinically investigated as potential therapeutic targets for treating PC [89]. An important consideration while using these antigens for immunotherapy could be their expression on normal cells [90]. Interestingly, TSA are generated due to specific somatic mutations in oncogenes, which are patient specific and could be used as personalized therapy [4]. Nonetheless, some of the immune checkpoint inhibitors, like anti-CTLA-4 and anti-PD-1, have shown partial success in PC immunotherapy [91,92]. Table 1 shows the current clinical trials on immunotherapy with multimodality treatments.

## 5. Conclusions

Overall amalgamated use of RT, HT, and immunotherapy is immensely promising as a tripartite modality for the treatment of PC, especially in overcoming the hurdles of an immunosuppressive PC TME and drawbacks of monotherapy. The most encouraging feature of a tripartite modality is that by employing HT in combination with immunotherapy or RT, the dose of these therapies could be optimized and can alleviate their side-effects without compromising their therapeutic benefits. Further, targeted HT can also inhibit the repair of RT-induced DNA damage in cancer cells, so a combined application of HT and immunotherapy could enhance the anti-cancer efficacy of RT by maximizing DNA damage and inhibiting the immunosuppressive TME. Furthermore, cells in the S-phase are also relatively radio-resistant because of extensive DNA repair capabilities, whereas they are more sensitive to HT. In PC, due to leaky tumor vasculature and low oxygen perfusion, tumor cells are more hypoxic and as a result exhibit a radioresistant phenotype. HT could minimize the anaerobic condition by increased oxygen supply due to increased blood circulation. These observations show the synergistic effect of combined use of RT, HT, and immunotherapy, and could pave the way for clinical translational research that could improve outcomes for PC patients. Currently, clinical research involving the tripartite treatment approach is lacking and needs further investigations to study the synergistic effect of tripartite treatment specifically, in addition to the therapeutic doses, time intervals, and sequence of these modalities to widen the therapeutic window and alleviate their individual toxic effect.

## Figures and Tables

**Figure 1 cancers-10-00469-f001:**
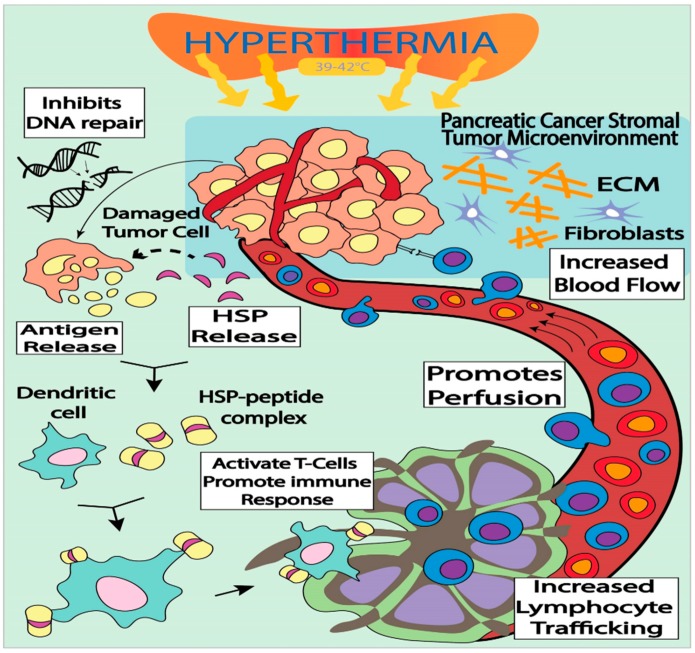
Integrated workflow of hyperthermia and its impact in activation of the immune response and inhibition of cancer cell DNA repair. Extra Cellular Matrix (ECM); Heat Shock Protein (HSP).

**Figure 2 cancers-10-00469-f002:**
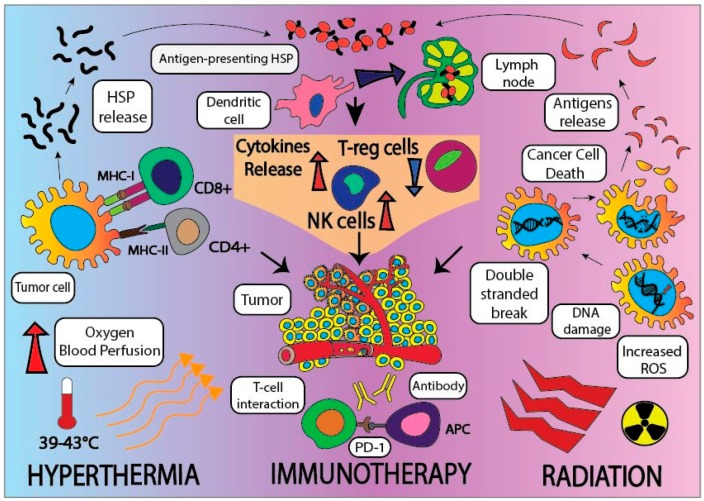
Schematic representation of tripartite modality combining hyperthermia with radiation and immunotherapy and their combined additive effect against cancer cells in the immunosuppressive tumor microenvironment (TME). NK cells: Natural Killer Cells.

**Table 1 cancers-10-00469-t001:** Pancreatic Cancer Clinical Trials with Hyperthermia.

Trials	Control Arm	Experimental Arm	Type of Hyperthermia
NCT01077427 Phase III	Adjuvant Gemcitabine and Capecitabine	Adjuvant Gemcitabine Cisplatin and regional hyperthermia	Regional hyperthermia
NCT02439593 Phase II	Chemo radiotherapy	Thermo chemo radiotherapy	Regional hyperthermia
NCT03251365 Phase II and III	Adjuvant chemotherapy Gemcitabine	HIPEC gemcitabine	Hyperthermic intraabdominal chemotherapy
NCT02862015 Phase II	Folfirinox or gemcitabine-based chemotherapy	Oncothermia	Whole body hyperthermia
NCT02973217	Standard chemotherapy	Specific form of thermotherapy—Immuno Stimulating interstitial laser thermotherapy	Thermotherapy

Note: Pancreatic cancer clinical trials employing hyperthermia with other modalities (radiation/chemotherapy/immunotherapy).

**Table 2 cancers-10-00469-t002:** Pancreatic Cancer Clinical Trials with Immunotherapy combined with RT and Chemotherapy.

Trials	Control Arm	Experimental Arm	Immunotherapy
NCT02405585 Phase II	Neoadjuvant chemotherapy followed by SBRT with Gemcitabine in borderline resectable pancreatic cancer	Neoadjuvant chemotherapy with Immunotherapy Algenpantucel-L Followed by SBRT with Gemcitabine	Algenpantucel-L
NCT01959672 Phase II	Neoadjuvant chemotherapy (gemcitabine, leucovorin, 5FU) followed by SBRT with Nelfinavir	Add Oregovomab with chemotherapy	Oregovomab (Chin)
NCT01072981 Phase III	Post-Surgery, adjuvant Gemcitabine, or 5FU chemo radiation	Post-Surgery, adjuvant Gemcitabine, or 5FU chemo radiation with Algenpantucel-L	Algenpantucel-L
NCT01903083 Phase I	No control	Chemo immunotherapy followed by assessment for surgery	Tadalafil
NCT02648282 Phase II	Chemotherapy with radiation therapy	GVAX vaccine and Pembrolizumab along with chemo radiation therapy	GVAX and Pembrolizumab
NCT03104439 Phase II	Radiation therapy	Nivolumab and Ipilimumab with radiation therapy	Nivolumab and Ipilimumab
NCT02305186 Phase II	Neoadjuvant chemoradiation	Neoadjuvant chemoradiation with Pembrolizumab	Pembrolizumab
NCT01342224 Phase I	No control	Vaccination with chemotherapy followed by radiation therapy	Tadalafil and Vaccination

Note: Ongoing or prior pancreatic cancer clinical trials employing immunotherapy with other modalities (radiation/chemotherapy therapy). SBRT: Stereotactic body radiation therapy, 5FU: Fluorouracil, GVAX: granulocyte-macrophage colony-stimulating factor (GM-CSF) gene-transfected tumor cell vaccine.

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
