# Peer review of "Immunotherapy, Radiotherapy, and Hyperthermia: A Combined Therapeutic Approach in Pancreatic Cancer Treatment"

_cancers, 2018, doi:10.3390/cancers10120469_

Round 1

Reviewer 1 Report

I believe this review has potential to be improved and can help enlighten Oncologists to consider including other strategies to approach this recalcitrant cancer besides just chemotherapy.

Several key references were not included.

If the review is primarily on hypothermia, there should be more on this topic including mechanism , animal studies etc and less on the general treatment of PC.

Specific comments below;

·         “Standard chemotherapeutic agents used to treat PC 57 include nucleoside analogs, such as gemcitabine.”

      Ø  Standard of care today uses FOLFIRINOX, this should be mentioned.

·         Reference 8 is not specifically about the stroma but about gemcitabine resistance. Perhaps referencing  Dr. Apte’s work from “Gastroenterology. 2013 Jun;144(6):1210-9.” Would be appropriate. Another PC cancer paper to reference is: “Feig C, Gopinathan A, Neesse A, Chan DS, Cook N, Tuveson DA. The pancreas cancer microenvironment. Clin Cancer Res 2012;18:4266-4276.

 ·         The introduction paragraph includes very well published and known material and can probably be deleted up to the sentence: “The recalcitrant nature of these tumors 66 is mostly due to the presence of a dense desmoplastic stroma,…../ The introduction can be about the chemoresistance of PC or emphasize what work has been done in hyperthermia and cancer therapy.

·         After this sentence “Recent investigations have revealed that immunotherapy when used in combination with other modalities show promising therapeutic outcomes for a variety of malignancies [11].” The reviewers should comment that immunotherapy has failed in PC (Brahmer JR, Tykodi SS, Chow LQ, Hwu WJ, Topalian SL, Hwu P, et al. Safety and activity of anti-PD-L1 antibody in patients with advanced cancer. N Engl J Med 2012;366:2455-65.)

·         In the section on combined agents with immune checkpoint antibodies, the authors show reference: “Cancer Immunol Immunother. 2018 Feb;67(2):195-207.”

·         The paper just needs a little organization. For example the paragraph on the Immune cells jumps back to talk about the Stroma.

·         Agents tried to decrease the stroma paragraph should include the trial with Hyaluronidase, that failed: “Clin Cancer Res. 2016 Jun 15;22(12):2848-54.”

·         Line 277 “RT has shown tremendous promise as a combined therapy to treat solid tumors in preclinical and clinical model.”… I think stating “tremendous” promise is an exaggeration since it has not cured anyone.

·   In the paragraph about vaccines and PD, the authors fail to reference 2 major trials using vaccines: Pancreas. 2012 Apr;41(3):374-9. (a vaccine to gastrin that was significant compared to placebo) and GVAX that failed in the clinic (J Clin Oncol 2015;33:1325-1333.)

Author Response

[Cancers] Manuscript ID: cancers-383422

Title: Immunotherapy, Radiotherapy, and Hyperthermia: A Combined Therapeutic Approach in Pancreatic Cancer Treatment.

Date: 11 Nov, 2018.

To the Editor,

We thank the reviewers for their very generous and complimentary comments on the review manuscript titled “Immunotherapy, Radiotherapy, and Hyperthermia: A Combined Therapeutic Approach in Pancreatic Cancer Treatment”.

As per the reviewer’s comments, we have incorporated the all necessary revisions in the manuscript and the manuscript is now fully revised for your kind consideration. All the changes have been made are highlighted in yellow to make it more easily readable.

Answers to the reviewers comments:

In the text; we addressed the concerns of Reviewer 1:

Specific comments; “If the review is primarily on hypothermia, there should be more on this topic including mechanism, animal studies etc and less on the general treatment of PC”. Several key references were not included.         

Response: We agree with the reviewer that our explanations were not adequate. This review is not primarily on hyperthermia, rather the combination of different treatment modalities to enhance radiation response to cure locally advanced unresectable pancreatic cancer. Hyperthermia enhances radiation and chemotherapy response is known for other cancers. Interestingly, there are accumulating evidences indicate that tumor specific targeted hyperthermia impacting the tumor microenvironment through temperature-sensitive check-points that regulate tumor vascular perfusion, lymphocyte trafficking, inflammatory cytokine expression, tumor metabolism, and innate and adaptive immune function. We highlighted in this review this novel combination (tripartite treatment) of radiation, hyperthermia and immunotherapy will open a new school of thoughts to treat pancreatic cancer more effectively.  We added more references as suggested by the reviewer.

Specific comments; “Standard chemotherapeutic agents used to treat PC include nucleoside analogs, such as gemcitabine”. Standard of care today uses FOLFIRINOX, this should be mentioned.

Response: (Line 57-58 added) include nucleoside analogs, such as gemcitabine or a combination drug therapy called FOLFIRINOX (infusional FU/leucovorin, oxaliplatin, and irinotecan).

Specific comments; “Reference 8 is not specifically about the stroma but about gemcitabine resistance. Perhaps referencing Dr. Apte’s work from “Gastroenterology. 2013 Jun;144(6):1210-9.” Would be appropriate. Another PC cancer paper to reference is: “Feig C, Gopinathan A, Neesse A, Chan DS, Cook N, Tuveson DA. The pancreas cancer microenvironment. Clin Cancer Res 2012;18:4266-4276”.

Response: (Line 433-438 added) Ref 8. Apte MV, Wilson JS, Lugea A, Pandol SJ. A starring role for stellate cells in the pancreatic cancer microenvironment. Gastroenterology. 2013 Jun;144(6):1210-9.

Ref 9. Feig C, Gopinathan A, Neesse A, Chan DS, Cook N, Tuveson DA. The pancreas cancer microenvironment. Clin Cancer Res. 2012 Aug 15;18(16):4266-76.

Specific comments; “The introduction paragraph includes very well published and known material and can probably be deleted up to the sentence: “The recalcitrant nature of these tumors is mostly due to the presence of a dense desmoplastic stroma,…../ The introduction can be about the chemoresistance of PC or emphasize what work has been done in hyperthermia and cancer therapy”.

Response: (Line 66-76 added) PC is more resistant to chemotherapy as well as to other standard treatment modalities compared to other solid tumors [7, 8]. Furthermore, the unique feature of PC is the presence of an abundant fibrous stroma which acts as a physical barrier to restrict intra-tumoral cytotoxic drug infiltration and creates a hypoxic microenvironment that decreases the efficacy of radiotherapy as well. PC is characterized by duct-like, tubular, highly fibrotic stromal growth, which is composed of extracellular matrix (ECM), stromal fibroblasts, various immunosuppressive cells, and a complex extracellular matrix comprised of glycosaminoglycans, proteoglycans and collagens. Hyaluronan (HA) is one of the predominant glycosaminoglycans present in the PC stroma. Recently, PEGylated human recombinant hyaluronidase combined with Gemcitabine tried to overcome the negative stromal environment, but ultimately failed to show benefit, highlighting the challenge to overcome the negative TME (9).

REF: (9) Hingorani SR, Harris WP, Beck JT, Berdov BA, Wagner SA, Pshevlotsky EM, Tjulandin SA, Gladkov OA, Holcombe RF, Korn R, Raghunand N, Dychter S, Jiang P, Shepard HM, Devoe CE. Phase Ib Study of PEGylated Recombinant Human Hyaluronidase and Gemcitabine in Patients with Advanced Pancreatic Cancer. Clin Cancer Res. 2016 Jun 15;22(12):2848-54.

Specific comments; “After this sentence “Recent investigations have revealed that immunotherapy when used in combination with other modalities show promising therapeutic outcomes for a variety of malignancies [11].” The reviewers should comment that immunotherapy has failed in PC (Brahmer JR, Tykodi SS, Chow LQ, Hwu WJ, Topalian SL, Hwu P, et al. Safety and activity of anti-PD-L1 antibody in patients with advanced cancer. N Engl J Med 2012;366:2455-65.)”.

Response: (Line 93-94 added) however, immunotherapy did not show any overwhelming effect in PC treatment outcome (REF: 13).

Specific comments; “In the section on combined agents with immune checkpoint antibodies, the authors show reference: “Cancer Immunol Immunother. 2018 Feb;67(2):195-207.”

Response: (Line 491-493 added)

REF: (28) Smith JP, Wang S3, Nadella S, Jablonski SA, Weiner LM. Cholecystokinin receptor antagonist alters pancreatic cancer microenvironment and increases efficacy of immune checkpoint antibody therapy in mice. Cancer Immunol Immunother. 2018 Feb;67(2):195-207.

Specific comments; “The paper just needs a little organization. For example the paragraph on the Immune cells jumps back to talk about the Stroma”.

Response: (Line 315-326 added) Hyperthermia releases of HSPs, enhancing the effect of IL-2 in activating T-lymphocytes and increases recruitment of tumor killing immune cells like natural killer cells, macrophages and cytotoxic and helper T-cells inside the tumor microenvironment. When hyperthermia is combined with radiotherapy the anti-tumor immune response would be augmented. Furthermore, the lack of clinical efficacy of PD-L1 blockade in pancreatic cancer patients suggests that it may be necessary to address the immunosuppressive effects by immune co-stimulatory agents (anti-CD134), hypofractionated RT or hyperthermia (HT). This tripartite treatment could reverse the immunosuppressive effect of the TME and from chemotherapy with an increased chemo sensitization effects exerted by RT and HT leading to tumor regression. The tripartite treatment will lead to the release of tumor antigens, recruit immune cells and improve the immunosuppression by chemotherapy in the tumor microenvironment and improving drug delivery by breaking the stromal barrier (74)

REF: (74) Miyamoto R, Oda T, Hashimoto S1, Kurokawa T, Inagaki Y, Shimomura O, Ohara Y, Yamada K, Akashi Y, Enomoto T, Kishimoto M, Yanagihara H, Kita E, Ohkohchi N. Cetuximab delivery and antitumor effects are enhanced by mild hyperthermia in a xenograft mouse model of pancreatic cancer. Cancer Sci. 2016 Apr;107(4):514-20.

Specific comments; “Agents tried to decrease the stroma paragraph should include the trial with Hyaluronidase, that failed: “Clin Cancer Res. 2016 Jun 15;22(12):2848-54.”

Response: (Line 66-76 added). PC is more resistant to chemotherapy as well as to other standard treatment modalities compared to other solid tumors [7, 8]. Furthermore, the unique feature of PC is the presence of an abundant fibrous stroma which acts as a physical barrier to restrict intra-tumoral cytotoxic drug infiltration and creates a hypoxic microenvironment that decreases the efficacy of radiotherapy as well. PC is characterized by duct-like, tubular, highly fibrotic stromal growth, which is composed of extracellular matrix (ECM), stromal fibroblasts, various immunosuppressive cells, and a complex extracellular matrix comprised of glycosaminoglycans, proteoglycans and collagens. Hyaluronan (HA) is one of the predominant glycosaminoglycans present in the PC stroma. Recently, PEGylated human recombinant hyaluronidase in combination with Gemcitabine showed promising therapeutic benefit in patients with advanced PDA, especially in those with high-HA tumors in a Phase 1b clinical trial (9).

REF: (9) Hingorani SR, Harris WP, Beck JT, Berdov BA, Wagner SA, Pshevlotsky EM, Tjulandin SA, Gladkov OA, Holcombe RF, Korn R, Raghunand N, Dychter S, Jiang P, Shepard HM, Devoe CE. Phase Ib Study of PEGylated Recombinant Human Hyaluronidase and Gemcitabine in Patients with Advanced Pancreatic Cancer. Clin Cancer Res. 2016 Jun 15;22(12):2848-54.

Specific comments; “RT has shown tremendous promise as a combined therapy to treat solid tumors in preclinical and clinical model.”… I think stating “tremendous” promise is an exaggeration since it has not cured anyone”.

Response: (Line 284) RT has shown promising therapeutic outcome (“tremendous” was deleted).

Specific comments; “In the paragraph about vaccines and PD, the authors fail to reference 2 major trials using vaccines: Pancreas. 2012 Apr;41(3):374-9. (a vaccine to gastrin that was significant compared to placebo) and GVAX that failed in the clinic (J Clin Oncol 2015;33:1325-1333.)”

Response: [Line 366 REF added]

REF: (85) Gilliam AD, Broome P, Topuzov EG, Garin AM, Pulay I, Humphreys J, Whitehead A, Takhar A, Rowlands BJ, Beckingham IJ. An international multicenter randomized controlled trial of G17DT in patients with pancreatic cancer. Pancreas. 2012 Apr;41(3):374-9.

REF: (86) Dung T. Le, Andrea Wang-Gillam, Vincent Picozzi, Tim F. Greten, Todd Crocenzi, Gregory Springett, Michael Morse, Herbert Zeh, Deirdre Cohen, Robert L. Fine, Beth Onners, Jennifer N. Uram, Daniel A. Laheru, Eric R. Lutz, Sara Solt, Aimee Luck Murphy, Justin Skoble, Ed Lemmens, John Grous, Thomas Dubensky, Jr, Dirk G. Brockstedt, and Elizabeth M. Jaffee. Safety and Survival with GVAX Pancreas Prime and Listeria Monocytogenes–Expressing Mesothelin (CRS-207) Boost Vaccines for Metastatic Pancreatic Cancer. J Clin Oncol. 2015 Apr 20; 33(12): 1325–1333.

Special Note: We figured out there was a small mistake in the Figure 2. We corrected the direction of the arrows in the immunotherapy box inside the figure, which is now corrected, revised, and replaced.

We are strongly hoping that this revised manuscript will be considered for publication in your esteemed journal “Cancers”.

 Sincerely yours,

 Javed Mahmood. PhD.

Reviewer 2 Report

The authors review the combination therapy with Immunotherapy, Radiotherapy and Hyperthermia against pancreatic cancer because this cancer type is therapy-resistant. This review well summarizes the effectiveness of the combination therapy and could provide interesting information to the readers of Cancers. However, some points must be clarified or corrected.

1) Page 3, line 111. The authors describe “IFN-g secreted by T cells triggers PD-1 upregulation (on cancer cells)”. PD-1 must be PD-L1.

2) Page 4, line 159. The authors describe “The overexpression --- VEGF, TNF-a and TGF-b also aid in cancer progression by inviting macrophages ---“. Is “inviting” correct? The author supposes that this must be “inhibiting infiltration of”.

3) Page 7, line 271. Checkpoint inhibitors such as CTLA-4 and PD-1 must be checkpoint inhibitors such as anti-CTLA-4 or anti-PD-1 antibody.

Author Response

[Cancers] Manuscript ID: cancers-383422

Title: Immunotherapy, Radiotherapy, and Hyperthermia: A Combined Therapeutic Approach in Pancreatic Cancer Treatment.

Date: 11 Nov, 2018.

To the Editor,

We thank the reviewers for their very generous and complimentary comments on the review manuscript titled “Immunotherapy, Radiotherapy, and Hyperthermia: A Combined Therapeutic Approach in Pancreatic Cancer Treatment”.

As per the reviewer’s comments, we have incorporated the all necessary revisions in the manuscript and the manuscript is now fully revised for your kind consideration. All the changes have been made are highlighted in yellow to make it more easily readable.

Answers to the reviewers comments:

In the text; we addressed the concerns of Reviewer 2:

Specific comments; Page 3, line 111. The authors describe “IFN-g secreted by T cells triggers PD-1 upregulation (on cancer cells)”. PD-1 must be PD-L1.

Response: (Line 119-120 added) (INF-γ) secreted by T cells triggers Programmed death-ligand 1 (PDL-1) upregulation and produces a PDL-1 signal.

Specific comments; Page 4, line 159. The authors describe “The overexpression --- VEGF, TNF-a and TGF-b also aid in cancer progression by inviting macrophages ---“. Is “inviting” correct? The author supposes that this must be “inhibiting infiltration of”.

Response: (Line 167-168 added) progression by inhibiting infiltration of macrophages. (Inviting deleted).

Specific comments; Page 7, line 271. Checkpoint inhibitors such as CTLA-4 and PD-1 must be checkpoint inhibitors such as anti-CTLA-4 or anti-PD-1 antibody.

Response:   (Line 278, 352) anti-CTLA-4 and anti-PD-1 added.                

Special Note: We figured out there was a small mistake in the Figure 2. We corrected the direction of the arrows in the immunotherapy box inside the figure, which is now corrected, revised, and replaced.

 We are strongly hoping that this revised manuscript will be considered for publication in your esteemed journal “Cancers”.

 Sincerely yours,

Javed Mahmood. PhD.

Reviewer 3 Report

In this manuscript, Mahmood and coworkers reviewed the combinational therapy of immunotherapy and radiation-induced loco-regional hyperthermia was able to activate a more robust immune response against the pancreatic cancer. The author also explained the mechanism of the enhanced anti-tumor response. It was due to the overexpression of the danger signal proteins such as heat shock proteins in response to the increased local temperature. The oxygenated hypoxic regions also contributed to the immunomodulatory effect at the tumor microenvironment. This is an interesting piece of work. The result revealed that the combinational system enhanced antitumor effects of immunotherapy significantly in animal models of various cancer types, with minimized systemic toxicity and side effects. This conclusion may provide alternatives for designing current drug delivery systems for immunotherapy. I suggest this manuscript could be accepted for publication after minor revision.

1. The authors provided very useful information of the ongoing clinical phase studies of the loco-regional hyperthermia along with other therapeutic modalities to boost the immunotherapy. This provided practical reference for the readers interested in hyperthermic therapy.

2. In Section 4, the authors provided “Vaccines against Pancreatic Cancer Neoantigen, and its success sin immunotherapy”. Since the maintext is about the combination of hyperthemia, immunotherapy and radiation, what is the reason to include this section in the manuscript?

3. The authors mentioned that the biggest challenges of treating PC are its dense stromal tissues and a highly immunosuppressive tumor microenvironment. In the later part of the manuscript, the author demonstrated that the treatment will reverse the immunosuppressive microenvironment. However, the author did not explain well on how the combinational treatment addresses the challenges of the stroma. More explanation or data are needed to support the statement.

4. From the translational perspective, a brief summary of FDA or EMA regulations for drug design and cell therapy to combine those treatment should be included, which could help the readers to find their research of interest.

5. Important reference about nanomaterials for drug delivery should be cited, such as: A melanin-mediated cancer immunotherapy patch 2017,  Polymeric Microneedles for Transdermal Protein Delivery, Advanced Drug Delivery Reviews 2018.

Author Response

[Cancers] Manuscript ID: cancers-383422

Title: Immunotherapy, Radiotherapy, and Hyperthermia: A Combined Therapeutic Approach in Pancreatic Cancer Treatment.

Date: 11 Nov, 2018.

To the Editor,

We thank the reviewers for their very generous and complimentary comments on the review manuscript titled “Immunotherapy, Radiotherapy, and Hyperthermia: A Combined Therapeutic Approach in Pancreatic Cancer Treatment”.

As per the reviewer’s comments, we have incorporated the all necessary revisions in the manuscript and the manuscript is now fully revised for your kind consideration. All the changes have been made are highlighted in yellow to make it more easily readable.

Answers to the reviewers comments:

In the text; we addressed the concerns of Reviewer 3:

Specific comments; “In Section 4, the authors provided “Vaccines against Pancreatic Cancer Neoantigen, and its success sin immunotherapy”. Since the maintext is about the combination of hyperthermia, immunotherapy and radiation, what is the reason to include this section in the manuscript?”.

Response: Recent several studies have shown that the more tumor-specific mutations, or neoantigens, the cancer cells have, the higher the chance that the tumor will not be endured by the immune system. Recent clinical trials are evaluating safety and immunogenicity of a neoantigen DNA vaccine strategy in pancreatic cancer patients following surgical resection and adjuvant chemotherapy. Pancreatic tumors are difficult to treat via immunotherapy because T cells don’t recognize them and therefore do not initiate an immune response.   There are evidences that neoantigens present in the tumors of the long-term survivors differed from those of other pancreatic cancer patients. This raises hope that neoantigens could be used as part of personalized cancer vaccines. It is very important to evaluate the effects of neoantigens with radiotherapy, chemotherapy, immunotherapy, and hyperthermia. Hence, we believe that this chapter will help to expand knowledge of the readers of “Cancers”.    

Specific comments; “The authors mentioned that the biggest challenges of treating PC are its dense stromal tissues and a highly immunosuppressive tumor microenvironment. In the later part of the manuscript, the author demonstrated that the treatment will reverse the immunosuppressive microenvironment. However, the author did not explain well on how the combinational treatment addresses the challenges of the stroma. More explanation or data are needed to support the statement.”

Response: (Lines 315-326 added) PC is more resistant to chemotherapy as well as to other standard treatment modalities compared to other solid tumors [7, 8]. Furthermore, the unique feature of PC is the presence of an abundant fibrous stroma which acts as a physical barrier to restrict intra-tumoral cytotoxic drug infiltration and creates a hypoxic microenvironment that decreases the efficacy of radiotherapy as well. PC is characterized by duct-like, tubular, highly fibrotic stromal growth, which is composed of extracellular matrix (ECM), stromal fibroblasts, various immunosuppressive cells, and a complex extracellular matrix comprised of glycosaminoglycans, proteoglycans and collagens. Hyaluronan (HA) is one of the predominant glycosaminoglycans present in the PC stroma. Recently, PEGylated human recombinant hyaluronidase in combination with Gemcitabine showed promising therapeutic benefit in patients with advanced PDA, especially in those with high-HA tumors in a Phase 1b clinical trial (74).

(74) Hingorani SR, Harris WP, Beck JT, Berdov BA, Wagner SA, Pshevlotsky EM, Tjulandin SA, Gladkov OA, Holcombe RF, Korn R, Raghunand N, Dychter S, Jiang P, Shepard HM, Devoe CE. Phase Ib Study of PEGylated Recombinant Human Hyaluronidase and Gemcitabine in Patients with Advanced Pancreatic Cancer. Clin Cancer Res. 2016 Jun 15;22(12):2848-54.

Specific comments; “From the translational perspective, a brief summary of FDA or EMA regulations for drug design and cell therapy to combine those treatment should be included, which could help the readers to find their research of interest”.

Response: (Lines 328-331 added) PC is usually marked by a poor prognosis with a significant morbidity and mortality. For deadly cancer like PC, an exciting and promising new therapy like tripartite treatment, if successful with relatively mild side effect in the future clinical trials in humans, may receive expedited approval from the European Medicines Agency (EMA) or the Food and Drug Association (FDA).

Specific comments; “Important reference about nanomaterials for drug delivery should be cited, such as: A melanin-mediated cancer immunotherapy patch 2017,  Polymeric Microneedles for Transdermal Protein Delivery, Advanced Drug Delivery Reviews 2018”.

Response:  (Lines 143 added) cancer vaccines, and immune stimulators such as cytokines, interleukins, or interferons [25, 28, 29, 30].

REF: (29) Ye Y, Wang C, Zhang X, Hu Q, Zhang Y, Liu Q, Wen D, Milligan J, Bellotti A, Huang L, Dotti G, Gu Z. A melanin-mediated cancer immunotherapy patch. Sci Immunol. 2017 Nov 10;2(17).

REF: (30) Ye Y, Yu J, Wen D, Kahkoska AR, Gu Z. Polymeric microneedles for transdermal protein delivery. Adv Drug Deliv Rev. 2018 Mar 1;127:106-118.

Special Note: We figured out there was a small mistake in the Figure 2. We corrected the direction of the arrows in the immunotherapy box inside the figure, which is now corrected, revised, and replaced.

We are strongly hoping that this revised manuscript will be considered for publication in your esteemed journal “Cancers”.

Sincerely yours,

Javed Mahmood. PhD.

Round 2

Reviewer 1 Report

References 85, 86 have been added bu they are in the incorrect location. They should be moved to the paragraph under tumor specific antigens Line 384 (I think). They do not fi where they currently are.Otherwise the paper is much improved and the authors have been responsive to the comments.

Author Response

Response: We agree with the reviewer that the References 85, 86 were in the incorrect location. They are moved to the paragraph under tumor-specific antigens Line 384.  The incorrect location is deleted and the corrected Ref is marked in green. We thank the reviewers for their very supportive comments.